# Synergistic Effect of Nilavembu Choornam–Gold Nanoparticles on Antibiotic-Resistant Bacterial Susceptibility and Contact Lens Contamination-Associated Infectious Pathogenicity

**DOI:** 10.3390/ijms25042115

**Published:** 2024-02-09

**Authors:** Essam S. Almutleb, Samivel Ramachandran, Adnan A. Khan, Gamal A. El-Hiti, Saud A. Alanazi

**Affiliations:** Cornea Research Chair, Department of Optometry, College of Applied Medical Sciences, King Saud University, Riyadh 11433, Saudi Arabia; esalsarhani@ksu.edu.sa (E.S.A.); adkhan@ksu.edu.sa (A.A.K.); gelhiti@ksu.edu.sa (G.A.E.-H.); saaalanazi@ksu.edu.sa (S.A.A.)

**Keywords:** contact lens contamination, microbial keratitis, antibiotic-resistant, Nilavembu Choornam gold nanoparticles, antimicrobial sustainability

## Abstract

Antibiotic-resistant bacterial colonies mitigate rapid biofilm formation and have complex cell wall fabrications, making it challenging to penetrate drugs across their biofilm barriers. The objective of this study was to investigate the antibacterial susceptibility of antibiotic-resistant bacteria and contact lens barrenness. Nilavembu Choornam–Gold Nanoparticles (NC–GNPs) were synthesized using NC polyherbal extract and characterized by UV-visible spectrophotometer, SEM-EDX, XRD, Zeta sizer, FTIR, and TEM analysis. Contact lenses with overnight cultures of antibiotic-resistant bacteria *K. pneumoniae* and *S. aureus* showed significant differences in growth, biofilm formation, and infection pathogenicity. The NC–GNPs were observed in terms of size (average size is 57.6 nm) and surface chemistry. A zone of inhibition was calculated for *K. pneumoniae* 18.8 ± 1.06, *S. aureus* 23.6 ± 1.15, *P. aeruginosa* 24.16 ± 0.87, and *E. faecalis* 24.5 ± 1.54 mm at 24 h of NC–GNPs alone treatment. In electron microscopy studies, NC–GNP-treated groups showed nuclear shrinkage, nuclear disintegration, degeneration of cell walls, and inhibited chromosomal division. In contrast, normal bacterial colonies had a higher number of cell divisions and routinely migrated toward cell multiplications. NC–GNPs exhibited antibacterial efficacy against antibiotic-resistant bacteria when compared to NC extract alone. We suggest that NC–GNPs are highly valuable to the population of hospitalized patients and other people to reduce the primary complications of contact lens contamination-oriented microbial infection and the therapeutic efficiency of antibiotic-resistant bacterial pathogenicity.

## 1. Introduction

The human cornea and ocular tissues are among the most sensitive organs in the body. They are constantly exposed to several risk factors, including allergens, debris, pathogens, desiccation, cosmetics, injury, and rubbing [1]. Recently, in the eye care field, the use of contact lenses (CLs) has become a safe, effective, and preferred method for correcting refractive errors or for cosmetic purposes [2]. In addition, CL wear also significantly enhances the risk of ocular complications like microbial keratitis, CL-related peripheral ulcer, cornea-conjunctival neovascularization, superior limbic keratoconjunctivitis, dry eye, and CL discomfort [3,4]. Studies report that wearing contact lenses is associated with changes in the ocular microbiota [5,6]. Compared with daily wear, overnight (extended) wear, using soft CLs or any other material, can lead to microbial keratitis and other corneal inflammatory complications [7,8]. Some other studies report that approximately 52–65% of new cases of microbial keratitis were observed to be associated with CL wearers, and the risk for microbial keratitis is 80 times greater for CL wearers compared to non–CL wearers [9,10]. The CL-associated microbial keratitis is caused by bacteria, particularly multidrug-resistant bacteria strains [11]. Ting et al., 2021 reports that CL-associated keratitis is caused by bacteria (91–100% of cases), fungi (3–7% of cases), and acanthamoebae (1–5% of cases) [12]. The main bacterial virulence infectious strains that cause bacterial keratitis include *Klebsiella* sp., *Staphylococcus aureus*, *S epidermidis*, *Pseudomonas aeruginosa*, and *Acinetobacter* sp. [13]. The clinical manifestation of bacterial keratitis includes redness of the eye, eyelid swelling, eye discomfort, and decreased sight [14]. Additionally, a few ocular infections also occurred due to the formation of bacterial biofilm, which develops antibiotic resistance to bacteria [15]. Moreover, CL-care solutions must have the potential to decrease the microbes and prevent biofilm formation on CL, which would eventually reduce the risk of CL-associated infections [16].

In recent years, the rate of microbial adhesion in contact lenses has been considered an important factor in serious eye problems, including microbial keratitis and red eye. In spite of early diagnosis and effective antibiotic therapy, a new treatment strategy is necessary to understand and acquire more potent prevention and treatment methods [17]. Contact-lens-related keratitis might change from small peripheral sterile infiltrates to infectious central ulcers with sight-threatening potential. Overnight wear and improper lens care are considered important causative factors for corneal infection [18]. Additionally, bacterial biofilm formation is highly resistant to many antimicrobials [19]. Studies also show that lens care solutions are effective against the growth of planktonic bacterial cells and are ineffective against bacterial biofilm [20,21]. In this study, we observed that CLs with overnight cultures of antibiotic-resistant bacterial strains *K. pneumoniae* and *S. aureus* showed significant differences in growth, biofilm formation, and infection pathogenicity. In contrast, non-resistant bacterial strains did not reveal any significant changes.

According to the estimates of the Center for Disease Control, each year, 2 million people are infected with drug-resistant microbes [22]. Also, another study showed that approximately 80% of ocular isolates of Methicillin-Resistant *Staphylococcus aureus* (MRSA) are highly resistant to the most commonly prescribed antibiotic, fluoroquinolones [23]. Moreover, the application of ophthalmic drugs also leads to variations in the drug delivery rates to the cornea, which limits the therapeutic efficacy [24]. Additionally, formulated synthetic drugs are risky for maintaining tear film and anterior epithelium cellular barriers, since active substances of reactive oxygen species (ROS) may adversely affect users of CLs. Thus, there is a search for a new ophthalmic drug delivery system that enhances drug residence time in the eye. The current study was undertaken to investigate a well-known natural medicinal polyherbal plants (Nilavembu Kudineer Choornam) extract formulated by gold nanoparticles, synthesized in an eco-friendly and cost-effective manner.

Plants with medicinal qualities have always been associated with the cultural behaviors and traditional knowledge of the people. In recent years, herbal medicine has acted as a promising strategy for various diseases, showing anti-analgesic, antioxidant, and anticancer with the least side effects [25]. Siddha medicine is the most ancient medical system of India and originated from South India. There are thirty-two Siddha internal medicines, and among them, two are recommended by the Indian Government to be used against viral infections, particularly Kabasura Kudineer Choornam for COVID-19 and swine flu, and Nilavembu Kudineer Choornam for dengue, chikungunya and COVID-19 [26,27]. Nine ingredients made up Nilavembu Choornam (NC; Table 1), among which is Andrographis paniculate main component [28]. It alleviates COVID-19 symptoms and is also used for the management of dengue and chikungunya due to its immunomodulating properties [28]. The ingredients of the NC decoction possess anti-inflammatory, antimicrobial, analgesic, antioxidant, antiviral, cytotoxic, hepatoprotective, and antidiabetic activities [29,30,31]. The present study was planned to evaluate the therapeutic drug efficacy of synthesized NC gold nanoparticles (NC–GNPs) and investigate the antibacterial susceptibility of an antibiotic-resistant microorganism and contact lens barrenness.

## 2. Results

### 2.1. Determination of Bacterial Infection and Biofilm Formation in Contact Lenses

Four innumerable bacterial strains of both antibiotic-resistant (*K. pneumoniae* and *S. aureus*) and non-resistant (*P. aeruginosa* and *E. faecalis*) were investigated to determine the bacterial characteristics that could be used to differentiate Gram-positive from Gram-negative bacteria (Figure 1A,B). In overnight cultures of *K. pneumoniae* and *S. aureus*, there was a significant difference in growth and microcolony formation (Figure 1A). In contrast, no significant difference was observed in non-resistant bacterial strains. Next, the cultured bacterial strains were adjusted to OD600 of 0.2 to serve as the baseline of a growth curve measurement to be incubated at different time intervals (Figure 1C). There was a significant difference between the cell counts of antibiotic-resistant bacteria and non-resistant bacteria at 12 to 32 h of incubation, with cell counts ranging from 0.1 to 0.2 OD600, depending on the time point. Antibiotic-resistant bacteria adhere to the cell wall, form colonies, and divide rapidly to recruit progenitor cells through continuous overlapping and gene manipulation.

Figure 1D,E represent the primary epithelial cells and corneal epithelium of bovine corneas co-cultured with *K. pneumoniae*, *S. aureus*, *P. aeruginosa*, and *E. faecalis* bacterial strains. The data show cellular pathogen aggregation and DNA fragmentation during overnight co-culture (Figure 1D). A morphology study indicates that overnight (16 h) incubation with *K. pneumoniae* and *S. aureus* exhibited over aggregations and induced cell death. The arrow marks indicate that antibiotic-resistant bacterial infections were distinguished by nuclear disintegration and cell necrosis (arrows shown in Figure 1E). Additionally, plasma membrane lysis was induced together with cell necrosis, which was due to a smaller number of microbes that were co-cultured in a time-dependent manner.

As shown in Figure 1F, the growth patterns for each organism were varied: micro-biofilm colony formation in *K. pneumoniae* and *S. aureus* progressed through an early and intermediate cellular phase, followed by extensive growth (16 h), which was indicated in propidium iodide (PI)-stained CLs. In comparison, bacterial growth and biofilm formation by *P. aeruginosa* and *E. faecalis* were reduced in growth up to 16 h. As four species reached the maturation phase at the time point of 16 h, biofilms grown up to 16 h were selected for subsequent experiments. Similarly, Figure 1G shows the colony forming with micro-biofilm formation. Scanning electron microscope analysis was performed to investigate the morphology of bacterial biofilm formed on CLs at 16 h of incubation, and data show biofilm morphology for the four bacterial species used (Figure 1G). Specifically, *K. pneumoniae* and *S. aureus* biofilm exhibited a dense network of cells that were arranged in multiple layers, forming microcolonies, with the surface of the CLs clearly visible (shown by arrows in Figure 1G). The *P. aeruginosa* and *E. faecalis* on the topographies of CLs observed an individual cellular morphology occurring mostly as monolayers without extracellular matrix aggregation.

### 2.2. Antibacterial Growth Inhibition of Nilavembu Choornam Decoction (NCt)

Figure 2A shows that the NC polyherbal decoction (NCt) had potent antimicrobial activity against both Gram-positive bacterial strains (*S. aureus* and *E. faecalis).* However, the Gram-negative antibiotic-resistant bacterial strain (*K. pneumoniae*) was more resistant when compared to the Gram-negative non-resistant bacterial strain (*P. aeruginosa*) growth inhibition. These antibacterial culture plate results illustrate that the 20 µL NCt zone of inhibition was denoted *K. pneumoniae* 6.4 ± 0.4, *S. aureus* 11.8 ± 0.91, *P. aeruginosa* 16.8 ± 1.31, *E. faecalis* 17.8 ± 1.39 mm at 24 h incubation. The antibacterial ability of NCt was compared to 20 µL standard Penicillin–Streptomycin solution (0.1% Pen–Strep). The zone of inhibition was represented for *K. pneumoniae* 23.8 ± 1.59, *S. aureus* 28.8 ± 0.90, *P. aeruginosa* 33.5 ± 1.5 and *E. faecalis* 32.8 ± 0.97 mm, respectively (Figure 2B). Among the four bacterial strains treated with Pen–Strep, *K. pneumoniae* and *S. aureus* proved to be more resistant, and its results showed a much smaller zone of inhibition.

Similarly, the bacterial growth kinetics were measured with control versus NCt and Pen–Strep incubated bacteria, as shown in Figure 2C,D, at different time intervals. In Figure 2C,D, antibiotic-resistant bacteria grew linearly until 16 h, then differed at 24 and 32 h. Despite 20 µL/mL NCt preserved all bacteria grew until 16 h, then decreased at 24 and 32 h. Incubation with Pen–Strep linearly decreased bacterial growth for all strains from 8 to 24 h (Figure 2D). It should be noted, in particular, that Pen–Strep 20 µL/mL incubation for 24 h reduced bacterial growth by more than 50% compared to threshold levels at 0 h. Consequently, this study was extended to synthesize and characterize NC–gold nanoparticles in the presence of antibiotic-resistant bacteria as well as microbial infections in contact lenses.

### 2.3. Synthesis of NC–GNPs

Using green synthesis, AuNPs can be synthesized in a nontoxic and environmentally safe manner. Plant-based extracts are frequently used in green synthesis methods, because their functional groups can reduce Au^3+^ to Au^0^. In our present study, the Nilavembu Choornam polyherbal extract (NCt) acts as a source of reducing agents for gold ions and also stabilizes the synthesized nanoparticles. In addition, Nilavembu Choornam extract (NCt) contains numerous antioxidants and polyphenolic compounds (Table 1) to stabilize and cap the gold nanoparticle synthesis. When slowly adding the NC extract (Figure 3A) to the gold ionic solution with continuous magnetic stirring, a color change occurs from yellowish brown to intense purple, which is clearly seen in Figure 3(A2). This indicates the reduction of gold ions and also confirms the gold nanoparticle synthesis. NCt of 2.5 mL produced a higher fine content of gold nanoparticles (Figure 3(A2)) compared to NCt of 5 and 10 mL (Figure 3(A1)).

### 2.4. Characterization of Synthesized NC–GNPs

UV-Vis absorption spectra indicate that NC extract and NC–GNPs have been optimized (Figure 3(B1,B2)). The UV-Vis absorption spectrum of NC–GNPs (Figure 3(B2)) displayed an intense peak at λmax = 534 nm, confirming the formation of a gold nanoparticle.

The spectra of the synthesized NC–GNPs are shown in Figure 3C. The representative FTIR analysis spectrum for the synthesized NC–GNPs exhibited peaks in the range from 400 to 4000 cm^−1^. It was found to be at 592, 1023, 1629, 2924, and 3433 cm^−1^, which could be primary amino, ester moiety, carbonyl, and aromatic compounds. The sharp peak at 1629 cm^−1^ was assigned to the C=O stretching vibration of amide I. Other sharp IR peaks at 1022 and 2923 cm^−1^ correspond to the C-H stretching vibration phenolic moiety. In addition, the FTIR wide sharp band at 3433 cm^−1^ was assigned to the O-H stretching vibration in the carboxy moiety.

An XRD pattern of the NC–GNPs was measured at the scan range of 5–90°. The crystalline sizes obtained by the X-ray diffractograms of the NC–GNPs supported the results of the TEM analysis. Ultimately, the XRD data for the NC–GNPs showed diffraction peaks at 2θ = 38.16°, 44.6°, 64.64°, 77.6°, and 81.74°, which can be indexed to the 1425, 441.66, 508.33, 566.66, and 241.66 reflection planes, confirming the face-centered cubic crystalline of nanogold. Additionally, the spectrum of XRD analysis confirmed the crystalline nature of the gold nanoparticles (Figure 3D).

Transmission electron microscopy (TEM) results are shown in Figure 4A,B. The synthesized NC–GNPs were spherical, almost homogeneous, and non-aggregated, with an average particle size of 57.6 nm (Figure 4A). The hydrodynamic size and stability of the NC–GNPs were assessed using DLS/zeta potential analysis. The average hydrodynamic size distribution analyzed by DLS was found to be 76.99 nm (Figure 4C). The values of the zeta potential (−20.9 mV in Figure 4C) clearly indicated that the negative charge of the surface nanoparticles might be due to the presence of biomolecules from the NC-polyherbal extracts, which provided good stability to synthesized NC–GNPs. Figure 4D SEM image clearly shows that the uniform distribution of the NC–GNPs EDX analysis demonstrated the presence of Au (62.68% mass detected at 2.12 keV) in NC–GNPs. A strong signal is indicated in the presence of an Au atom at 2.12 keV (Figure 4D).

### 2.5. Growth Inhibition of NC–GNPs in Antibiotic Resistance Bacterial Strains

The synthesized NC–GNPs were purified and diluted with distilled water after quantification. It was shown in Figure 5A,B that the NC–GNPs had potent antimicrobial activity against both Gram-negative and Gram-positive bacterial strains in a concentration manners (5, 10, 15, and 20 µg/vol). The culture plate results illustrate that the 20 µg/vol NC–GNPs zone of inhibition was denoted *K. pneumoniae* 19 ± 0.98, *S. aureus* 21 ± 1.04, *P. aeruginosa* 21.6 ± 1.03 and *E. faecalis* 23.2 ± 0.80 mm at 24 h of incubation, respectively. Among the four concentrations of NC–GNP treatment with bacterial strains, 20 µg/vol provided more zone of inhibition in all the bacterial strains. Similarly, the bacterial growth inhibition was plated with NC–GNPs (20 µg/vol) versus Pen–Strep incubation, as shown in Figure 5C,D at 24 h incubation. It should be noted, in particular, that NC–GNPs (*K. pneumoniae* 18.8 ± 1.06, *S. aureus* 23.6 ± 1.15, *P. aeruginosa* 24.16 ± 0.87, and *E. faecalis* 24.5 ± 1.54 mm) and Pen–Strep (*K. pneumoniae* 24.6 ± 0.81, *S. aureus* 30.0 ± 1.37, *P. aeruginosa* 32.4 ± 1.63 and *E. faecalis* 32.8 ± 1.39 mm) incubation for 24 h reduced the bacterial zone of inhibition by more than 70% compared to threshold levels at control.

### 2.6. Ultrastructural and Cellular Degeneration of NC–GNPs

A study was conducted to determine the impact of NC–GNPs on antibiotic susceptibility in Gram-negative and Gram-positive bacteria. TEM microphotographs of the investigated bacterial strains after incubation in the NC–GNPs suspension were distinguished in Figure 6A–H. Microphotograph results demonstrate the cell walls of Gram-positive bacterial strains (*S. aureus* and *E. faecalis*; Figure 6D,H) highly aggregates in NC–GNPs when compared with the Gram-negative bacterial strains (*K. pneumoniae* and *P. aeruginosa*; Figure 6B,F). The cross-sections of single resin-embedded cells of the studied bacteria illustrate the different shapes of the bacterial cells and suggest that the nanoparticles do not penetrate the Gram-positive bacterial cells. However, in Gram-negative bacterial cells, smaller particles directly penetrate the cell wall membrane and increase the degeneration of the nuclear bodies when compared to control bacteria. Further, DLS-Zita potential results showed that the NC–GNPs had strong negative charges, and those Au nanoparticles selectively adhered to Gram-positive bacterial cell membrane surfaces when compared to Gram-negative bacterial strains. Moreover, NC–GNP treated bacteria exhibited nuclear shrinkage, nuclear disintegration, degeneration of cell walls, and inhibited chromosome replication. In contrast, control bacterial colonies (Figure 6A,C,E,G) had a higher number of cell divisions, and they routinely migrated towards cell replications (C and D stages) than NC–GNP-treated bacteria.

## 3. Discussion

Our analysis of corneal epithelial cells co-cultured with antibiotic-resistant and non-resistant bacterial strains showed nuclear disintegration and cell necrosis. Studies reported that 80% of corneal ulcers are caused by *S. aureus*, *S. pneumoniae*, and *Pseudomonas* species. Also, they showed that early diagnosis is necessary to minimize the possibility of vision loss and to decrease corneal damage [32,33]. SEM analyses are performed to characterize the surface topography and 3D architecture of bacterial biofilms. Our experimental results, SEM analysis data, show that the *K. pneumoniae* and *S. aureus* biofilm exhibited microcolonies in multiple layers on CLs, while *P. aeruginosa* and *E. faecalis* form monolayers without extracellular matrix on CLs. Consistent with our study, Szczotka-Flynn et al., 2009 reported that bacterial (*P. aeruginosa*, *S. marcescens*, and *S. aureus*) biofilm were formed on a Silicon hydrogel CLs, which was inhibited in the presence of tested lens care solutions, while biguanide preserved care solutions were not effective against bacterial biofilms in vitro model [20]. Several studies used SEM analysis to understand bacterial adhesion and biofilm formation on CLs [32,33,34]. Consistent with our results, a cluster of microcolonies was reported using SEM analysis, and biofilm formation on CLs was evaluated (around 55%). Additionally, in our study, we observed that biofilm formation on contact lenses by different bacteria enters into distinct developmental phases. Szczotka-Flynn et al., 2009 reported that biofilm formation by *P. Aeruginosa*, *S. marcescens*, and *S. aureus* reached the mature phase by 18 h with a difference in growth kinetics [20]. In another study, it was reported that differential growth kinetics of biofilm formation by *S. epidermidis* and *E. faecalis* occurs due to change in hydrodynamic flow and exhibits distinct latent, accelerated, linear growth, and a mature phase upon biofilm formation [35].

As further investigation, we conducted the synthesis and characterization of gold nanoparticles with NC, which showed an intense peak at 534 nm. It maintained its stability, which suggests that further growth agglomeration of prepared nanoparticles would not have occurred, which is similar to that of other studies showing a distinct peak at 529 nm [36]. NC–GNP diffraction patterns showed sharp reflections at NC–GNP peaks, confirming the face-centered cubic crystalline (fcc) structure of nanogold.. Similarly, according to the XRD data of Krishnamurthy et al., 2014, XRD diffraction of gold nanoparticles was found to be at 2θ = 38.1, 44.3, 64.5, and 77.7, confirming the fcc crystal nature of gold nanoparticles [37]. FTIR analysis showed various peaks at 591, 1022, 1629, 2923, and 3432 cm^−1^, which could be primary amino, ester moiety, carbonyl, and aromatic compounds. The change in two peaks at 3432 and 1629 cm^−1^ indicated the reduction of gold ions into gold atoms. Studies showed that the presence of phenolic, alcoholic, and carboxylic compounds are essential for the reduction and stabilization of gold nanoparticles [38,39]. The TEM analysis showed the dispersity of the synthesized gold nanoparticle with the size of 57.6 nm (non-aggregated), which was similar to that of Lata et al., 2014 who reported that the TEM image of a gold nanoparticle showed the presence of spherical dark colored dot with monodispersed state with diameter 35 nm [40]. DLS/zeta potential analysis showed the average hydrodynamic size distribution as 76.99 nm for NC–gold nanoparticles. The difference in particle size with TEM and DLS analysis showed that DLS analysis includes the thickness of bio-compounds that are present on the surface of synthesized gold nanoparticles [38]. Similar to our data on energy dispersive X-ray analysis, Elavazhagan and Arunachalam, 2011 showed an optical absorption band peak at approximately 2.2 keV, which indicated the absorption of metallic gold nano-crystallites [41].

The antibacterial study results indicated that Nilavembu Choornam polyherbal decoction (NCt; 20 µL/disc) exhibited less antimicrobial activity against antibiotic-resistant bacteria and had more resistance when compared to the non-resistant bacterial growth inhibition. In a previous study it was reported that a Nilavembu Kudineer capsule (200 mg/kg) has very effective antimicrobial, analgesic, and antipyretic potential, exhibiting a higher level of zone of inhibition against *S. aureus*, *S. flexneri*, *P. aeruginosa*, *P. vulgaris*, and *K. pneumonia*, together with decreased inflammatory markers, including cyclooxygenase and prostaglandins, and also reduced proinflammatory markers IL-6 and IL-1 [42]. Studies showed that the presence of the peptidoglycan layer in Gram-positive bacteria makes them more permeable to plant extracts [43,44]. Apart from antibacterial effects, Nivalembu Kudineer (NVK) has also been reported to protect against dengue and the chikungunya virus, thereby inhibiting viral infection [36]. Moreover, NVK also exhibited a statistically significant reduction in hospital stay time and a reduction in viral load of SARS-CoV-2 among patients [37]. However, in our study, we found that NC maintained bacterial growth till 16 h. It was decreased at 24 and 32 h, while Pen–Strep incubation exhibited linear growth decrease from 8 to 24 h; thus, to improve the effectiveness of NCt, gold nanoparticle was synthesized using NC and the effect of NC–gold nanoparticle against antibiotic-resistant bacteria was investigated.

Further study showed that the optimum 20 µg/disc NC–GNPs had potent antimicrobial activity against both Gram-negative and Gram-positive bacteria. In a study, silver nanoparticle synthesis using Nilavembu (*Andrographis paniculata*) effectively inhibited the growth of both Gram-positive and Gram-negative bacteria *S. aureus* (17 mm), *K. pneumonia* (22 mm), and *S. typhimurium* (27 mm) at 1000 µg/disc, and also exhibited antifungal *C. albicans* (8 mm) and *Botrytis* (17 mm) [45]. Further, in our experiment, the ultrastructure of NC–GNPs treated bacteria exhibited nuclear shrinkage, nuclear disintegration, degeneration of cell walls, and inhibited chromosome replication, precisely in Gram-negative bacteria when compared to Gram-positive bacteria due to the presence of cell walls. Studies showed that pure gold nanoparticles exert antibacterial activity against both Gram-positive and Gram-negative strains by targeting their cytoplasmic membrane integrity [46]. In another study, it was reported that the weakening of DNA replication, damaging cell membranes, and inactivating proteins are considered important mechanisms behind the antibacterial activity of nanoparticles [47]. Additionally, in our study, we found that Gram-positive bacterial strains’ (*S. aureus* and *E. faecalis*) cell walls adhered much more with NC–GNPs when compared to the Gram-negative bacterial strains. Studies reported that gold nanoparticles are agglomerated on the bacterial surface and interact with the membrane protein due to their affinity towards proteins, which disrupts the bacterial membrane and breaks the interaction with cytoplasmic proteins, leading to cell death, and which involves another mechanism of antibacterial activity of NC–gold nanoparticle [48].

To our knowledge, this is the first study to evaluate and synthesize gold nanoparticles from NC polyherbal plant extract. The collective study results indicate that the NC–GNP’s treatment inhibits microbial adhesion and biofilm formation in CLs. NC–GNPs also reduce microbial infection and adhesion in CLs (Appendix A), although no study was conducted on normal corneal epithelium in vitro or in vivo. We performed a preliminary study regarding the safety and biocompatibility of NC–GNPs to the ocular and contact lens barrier functions in our ongoing project of normal epithelial cells representing NC–GNPs. The efficacy of gold nanoparticles, as well as NCt, is well known in the field of antibacterial sustainability, anticancer, and other biological applications. Therefore, future studies should examine the long-term biocompatibility and potential toxicity of GNPs in an ocular environment.

## 4. Materials and Methods

### 4.1. Chemicals

Nilavembu Choornam powder (NC), marketed by a commercial manufacturer, was procured from the Siddha storekeeper at Villupuram, Tamil Nadu, India. The hydrogen tetrachloroaurate (III) hydrate (HAuCl_4_.3H_2_O; Sigma Aldrich, St. Louis, MO, USA), syringe filtrate membrane (Sartorius Stedim Biotech, Göttingen, Germany), and other reagents were procured. For transmission electron microscopy (TEM) analysis, carbon-coated copper grids (TABB Bioscience (UK), Luria Bertani (LB), Nutrient, Brain Heart Infusion (BHI), and MacConkey broth/agar culture media (Scharlab S.L., Barcelona, Spain) were also procured.

### 4.2. Bacterial Strains and Contact Lenses

Bacterial strains, including *Klebsiella pneumoniae* (ATCC-700603), *Streptococcus aureus* (MRSA; ATCC-43300), *Pseudomonas aeruginosa* (ATCC-27582), and *Enterococcus faecalis* (ATCC-29212) were used, and Department of Microbiology, King Saud University provided specialties. The selected bacterial species are considered common causative agents of contact lens-related infection. According to the FDA recommendations and Premarket Notification 510(k) Guidance Document for Contact Lens Care Products, the American Type Culture Collection (ATCC) bacterial strains were used in the study. Bacteria were cultured overnight at 37 °C in different broths (Scharlab S.L., Spain) and washed thrice with PBS, and the obtained cell suspension was corrected spectrophotometrically at wavelength 600 nm. Avaira soft contact lenses (CooperVision, Pleasanton, CA, USA) were selected for the current research.

### 4.3. Biofilm Formation and Quantification

The contact lens was washed with PBS and incubated in twelve-well plates containing 4 mL of bacterial cell suspensions (0.2 OD at 600 nm) at different time intervals, including 4 h, 8 h, and overnight (16 h), at 37 °C for the adhesion of bacterial colonies onto the lens surface (adherence phase). After incubation and washing with PBS, the lens with bacterial biofilm adhered was transferred into a conical tube containing PBS. To quantify biofilms, the propidium iodide fluorescence staining was performed on lenses in which biofilms were allowed to form using different bacterial strains. The images were taken with a BX53 fluorescence microscope equipped with a CCD camera (Olympus, Tokyo, Japan), and the Cell Sensentry software 2.0 (Olympus, Tokyo, Japan) was used to analyze images.

### 4.4. Assessment of Bacterial Infection in Bovine Cornea and Corneal Epithelial Cells

Bovine eyes were separated from a local abattoir within 4 h post-mortem. The globe lens with the ciliary body was removed, and the cornea with a rim of sclera was collected and washed with PBS solution containing antibiotics. The digestion and epithelial cell isolations were performed to collect cells from distinct corneal layers, and the previous method was followed [49]. Separate sets of corneoscleral rings and epithelial cells were placed into six-well plates containing DMEM medium suspended with bacterial cells (OD was 0.2 at 600 nm) and were incubated overnight for 16 h at 37 °C in a humidified 5% CO_2_ incubator to allow adhesion of bacterial colonies on the epithelium surface. After the incubation, the corneas were fixed in 4% buffered formalin. Then followed H&E staining, and epithelial cells followed PI staining and were assessed under Olympus BX51 bright field and BX53 fluorescence microscopy, and further analyzed using Cell Sensentry software 2.0 (Olympus, Tokyo, Japan).

### 4.5. Scanning Electron Microscopy (SEM) Analysis

SEM analysis was performed to understand the surface topography of bacteria grown as biofilms. Briefly, *K. pneumoniae*, *S. aureus*, *P. aeruginosa*, and *E. faecalis* were allowed to develop biofilms on the CLs and attain a mature phase (16 h). Later, the contact lenses with mature biofilms were fixed with 2.5% glutaraldehyde and treated with 1% osmium tetraoxide, 2% tannic acid, and uranyl acetate with a sequence of ethanol dehydration process. Then, the prepared samples were sputter-coated with Au-Pd (60/40 ratio) and viewed under a microscope (JSM-6360, JEOL, Tokyo, Japan).

### 4.6. Preparation and Synthesis of Nilavembu Choornam Extracts (NCt) and NC–GNPs

Nilavembu Choornam powder consists of a polyherbal formulation with equal quantities of 9 herbs, which are listed in Table 2. Here, 1.5 gm of NC dried powder was added to 200 mL of distilled water and incubated for twenty minutes in a boiling water bath. Next, the extract was cooled and filtered through Whatman No. 1 filter paper and a filter membrane (0.45 µm). The filtrate was further used for the synthesis of gold nanoparticles.

Initially, 10 mL of the above concentrated NC filtrate was mixed with 40 mL of HAuCl_4_·3H_2_O (0.1 mM) solution, and later, a color change was observed in the reaction mixture with continuous magnetic stirring, which indicated the formation of gold nanoparticles (AuNPs). Next, the different volumes of diluted NC extract were then used to obtain optimized nanoparticles at 37 °C. To obtain purified AuNPs, the reaction mixture was centrifuged at 10,000× *g* for 10 min, and then the final residue was collected, dried, and stored. The obtained pure AuNPs were subject to characterization and used for further experimental studies. The formation of AuNPs was confirmed via an ultraviolet-visible (UV-Vis) spectrophotometer (Genesys 10S, Waltham, MA, USA) by measuring the absorbance at 200–800 nm. A control experiment without adding NC extract was also performed.

### 4.7. Characterization of NC–GNPs

Synthesized NC–GNPs were characterized by ultraviolet–visible Spectroscopy (Genesys 10S, Thermo Scientific, Waltham, MA, USA). In a 2 mL cuvette, NC–GNPs were scanned over a range of wavelengths from 200 to 800 nm to determine the peak value of absorbance. Fourier transform infrared spectroscopy (FTIR) analysis was performed, mixing 100 mg of potassium bromide powder with 2 mg of the NC–GNPs using a pelletizer to develop thin discs. The obtained disc sample was analyzed (400 to 4000 cm^−1^) using FTIR spectroscopy (Perkin Elmer RX-I, Shelton, CT, USA) with a resolution of 1 cm^−1^. The XRD pattern of the nanoparticle was measured (step size of 2°/min) using X’Pertt Pro X-ray diffractometer (Panalytical, Malvern, UK) with current and voltage settings as 30 mA and 40 kV, respectively, at the 2θ range of 10°–80° using CuKα X-ray source (λ = 1.5406 Å). The particle size distribution of the nanoparticle was evaluated using a particle size analyzer (NANOPHOX, Attica, Greece). Scanning electron microscopy (SEM; JSM-6360, JEOL, Welwyn Garden City, UK) was used at 7.50 kV to perform the morphological investigation by vacuum drying a drop of NPs on a graphite grid. Moreover, the elements present in the nanoparticle were micro-analyzed on the SEM with an energy dispersive X-ray microanalyzer (EDS, Oxford INCA, Gujarat, India). The morphology of gold nanoparticles and transmission electron microscopes (HT-1400, JEOL, Tokyo, Japan) were used with corban-coated grids. The nanoparticle was ultrasonicated in acetone and placed on a copper grid covered with carbon.

### 4.8. Antimicrobial Sustainability Assay

The antimicrobial activity of both NC extract and synthesized NC–GNPs was evaluated against four different bacterial strains (Gram-positive bacteria (*S. aureus* and *E. faecalis*) and Gram-negative bacteria (*K. pneumoniae* and *P. aeruginosa*)) by the agar well diffusion method. Bacterial strains were grown in 10 mL of different nutrient broth and cultured in a shaking incubator for 16 h at 37 °C to achieve 1 × 10^7^ CFU turbidity. Sterilized inoculation loops were used to distribute cultures of the above strain over medium plates. Upon spreading, 6 mm sterilized cotton discs were used to bore incubation. Then, 20 μL of each prepared NC extract and NC–GNP suspension at various concentrations (5, 10, and 20 μg/vol) was placed into separate discs and allowed to culture at room temperature. Also, 20 μL of 0.1% Penicillin–Streptomycin were used as positive controls. For 24 h, the bacterial plates were placed at 37 °C, and the diameter of the growth inhibition zone was evaluated after a suitable incubation time.

### 4.9. Transmission Electron Microscopy Analysis

Antibiotic-resistant bacterial strains of *K. pneumoniae* and *S. aureus* and normal bacterial strains *P. aeruginosa* and *E. faecalis* were grown overnight (16–18 h) in 5 mL different nutrient broth medium at 37 °C and adjusted to an optical density equal to 0.5 at 600 nm the day before testing. Next, the organism was transferred to the test tube with a medium and then allowed to reach 10^7^ colonies before the NC–GNPs treatments. 20 µL/mL of NC–GNP solutions were added to each test tube and incubated at 37 °C for 24 h. The next day, the nutrient broth medium suspended with bacterial colonies was washed with PBS, and fixation was completed with 2.5% glutaraldehyde in 0.1 M phosphate buffer for 2 h and washed with PBS. The cells were then fixed with 1% osmium tetroxide for 1 h, dehydrated in a series of alcohol, and embedded in epoxy resin. Samples were then allowed to polymerize in a 60 °C oven for 24 h. Thin sections were made at 80 nm thickness and viewed through JEOL HT-1400 transmission electron microscopy at 120 kV.

### 4.10. Statistical Analysis

The obtained data were used for calculating Mean ± Standard Error (SE), and statistical analysis was performed by GraphPad Prism 8.2 tools (GraphPad Software, San Diego, CA, USA). The datasets were analyzed using the Mann–Whitney unpaired *T*-test. The *p* values obtained through the Mann–Whitney unpaired *T*-test and multiple comparison tests that were less than 0.05, 0.01, and 0.001 are represented by *, **, and *** to indicate statistical significance. A triplicate of each experiment was performed in this study.

## 5. Conclusions

The polyherbal plant extract of Nilavembu Choornam was successfully used to synthesize gold nanoparticles through a plant-mediated green synthesis method. NCt was found to contain various phytoconstituents that were responsible for the formation and stability of Au NPs. Analysis of UV-Vis spectroscopy, FTIR, TEM, SEM-EDX, and XRD results revealed the size, morphology, crystalline structure, and stability of the NC–GNPs. Compared to NC extract, biosynthesized NC–gold nanoparticles had effective antibacterial effects on both antibiotic-resistant Gram-positive and Gram-negative bacteria. It appears that gold nanoparticles attenuate the adhesion and formation of bacterial biofilms on contact lenses. Hence, our present research results suggest that the navel therapeutic efficacy of NC–GNPs may contribute to better drug efficacy and antibiotic resistance to bacterial decontamination.

## Figures and Tables

**Figure 1 ijms-25-02115-f001:**
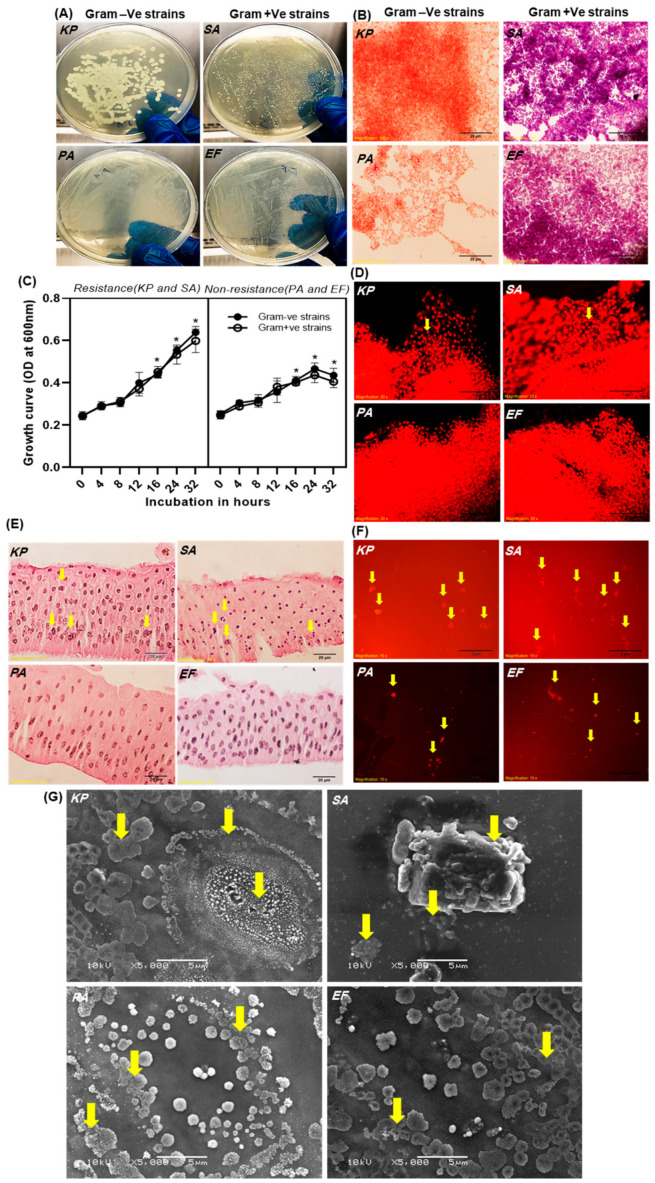
Assessment of bacterial infections, adhesion, and biofilm formation on the contact lenses: (**A**) schematic illustration of the LB agar plates cultured with different bacterial strains *K. pneumoniae*, *S. aureus*, *P. aeruginosa*, and *E. faecalis*; (**B**) light micrograph of Gram-iodide stained Gram-positive and Gram-negative bacterial strains (100×), scale bar = 20 μm; (**C**) graph differentiating the growth kinetics of antibiotic-resistance and non-resistance bacterial strains at different time intervals (0–32 h). * represent significant compared to control, *p* ≥ 0.05; (**D**) fluorescent microscopic images of primary cultured bovine corneal epithelial cells incubated with *K. pneumoniae*, *S. aureus*, *P. aeruginosa*, and *E. faecalis* at 16 h (20×). The arrows indicate nuclear fragmentation and cell damage, scale bar = 1 μm; (**E**) light microscopic images of co-cultured bovine corneal epithelium incubated with *K. pneumoniae*, *S. aureus*, *P. aeruginosa*, and *E. faecalis* stained by H&E (60×). The arrows show cellular pathogen aggregation and DNA fragmentation in basal epithelial cells, scale bar = 20 μm; (**F**) fluorescent microscopic images of contact lenses incubated with *K. pneumoniae*, *S. aureus*, *P. aeruginosa*, and *E. faecalis* (10×). The arrows indicate dense bacterial colonies adhering to each other and forming biofilms in CLs, scale bar = 2 μm; and (**G**) scanning electron micrograph of contact lens after being immersed and incubated with *K. pneumoniae*, *S. aureus*, *P. aeruginosa*, and *E. faecalis* at 16 h, scale bar = 5 μm. The arrows indicate dense bacterial colonies adhering to each other and forming biofilms in CLs.

**Figure 2 ijms-25-02115-f002:**
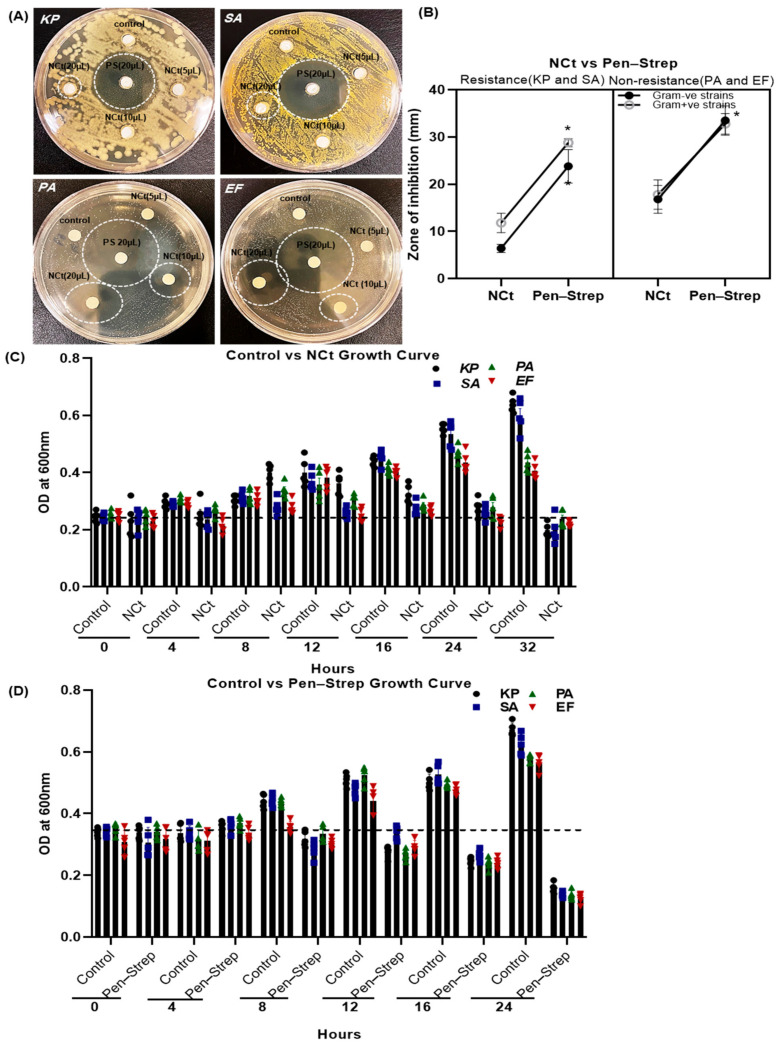
Antimicrobial and bacterial growth inhibition activity of Nilavembu Choornam decoction (NCt): (**A**) antimicrobial activity of Nilavembu polyherbal decoction (NCt) and Penicillin–Streptomycin (Pen–Strep) against antibiotic-resistant (*K. pneumoniae* and *S. aureus*) and non-resistant (*P. aeruginosa*, and *E. faecalis*) bacterial zone inhibitions. White dotted circles indicate the inhibition zone of NCt and Pen–Strep treatments; (**B**) graph showing the (**A**) zone of inhibition of NCt and Pen–Strep on the bacteria *K. pneumoniae*, *S. aureus*, *P. aeruginosa*, and *E. faecalis*. * represent significant compared to control, *p* ≥ 0.05. In plating, Luria Bertani (LB), Nutrient, and Brain Heart Infusion (BHI) agars were used; and (**C**,**D**) graphs representing OD600 bacterial growth kinetics of control bacteria versus bacteria treated with NCt and Pen–Strep at different time intervals (0–32 h). Luria Bertani (LB), Nutrient, and Brain Heart Infusion (BHI) broths were used for bacterial incubation. Also, 10 µL of distilled water was added to the control sample disc.

**Figure 3 ijms-25-02115-f003:**
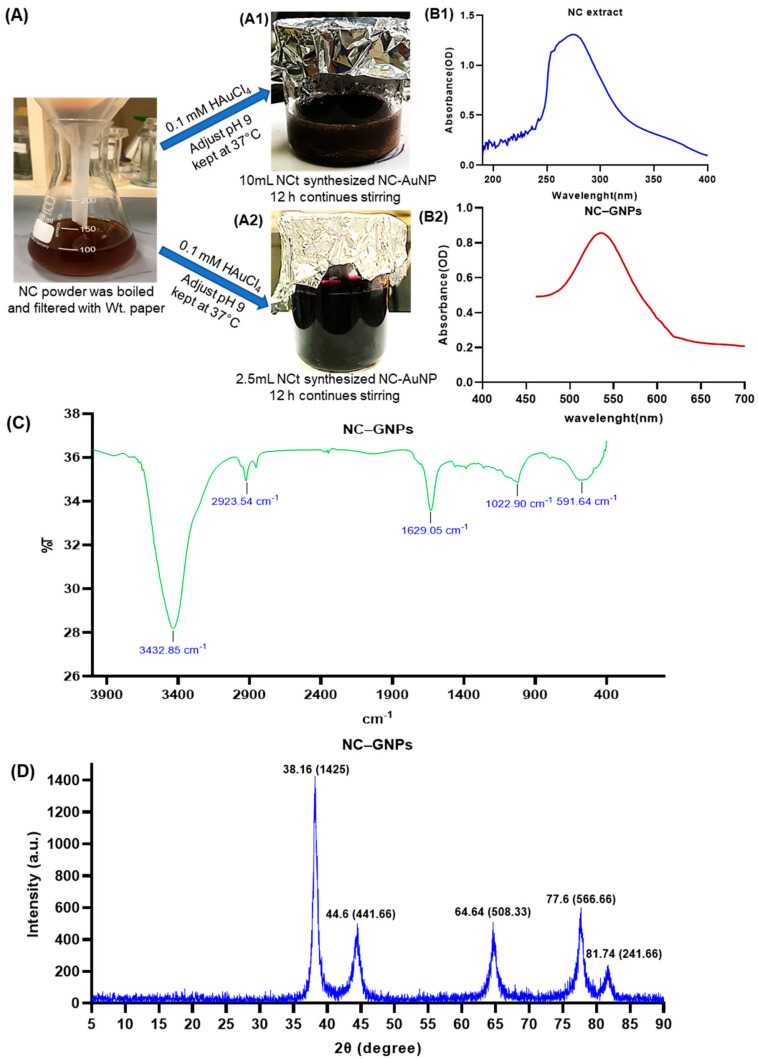
Synthesis and functional characterization of NC–gold nanoparticles: (**A**) shows NC powder extract was boiled and filtered decoction. (**A1**,**A2**) the optimization of NC extracts using different volumes; (**B1**,**B2**) the UV-Vis absorption spectral analysis of optimized NCt and NC–GNPs, (**B2**) spectrum displayed an intense peak at λmax = 534 nm confirming the formation of a gold high transparent solution; (**C**) graph showing Fourier transform infrared spectra (FTIR) of NC–GNPs; and (**D**) graph showing the X-ray diffraction (XRD) spectrum of synthesized NC–GNPs.

**Figure 4 ijms-25-02115-f004:**
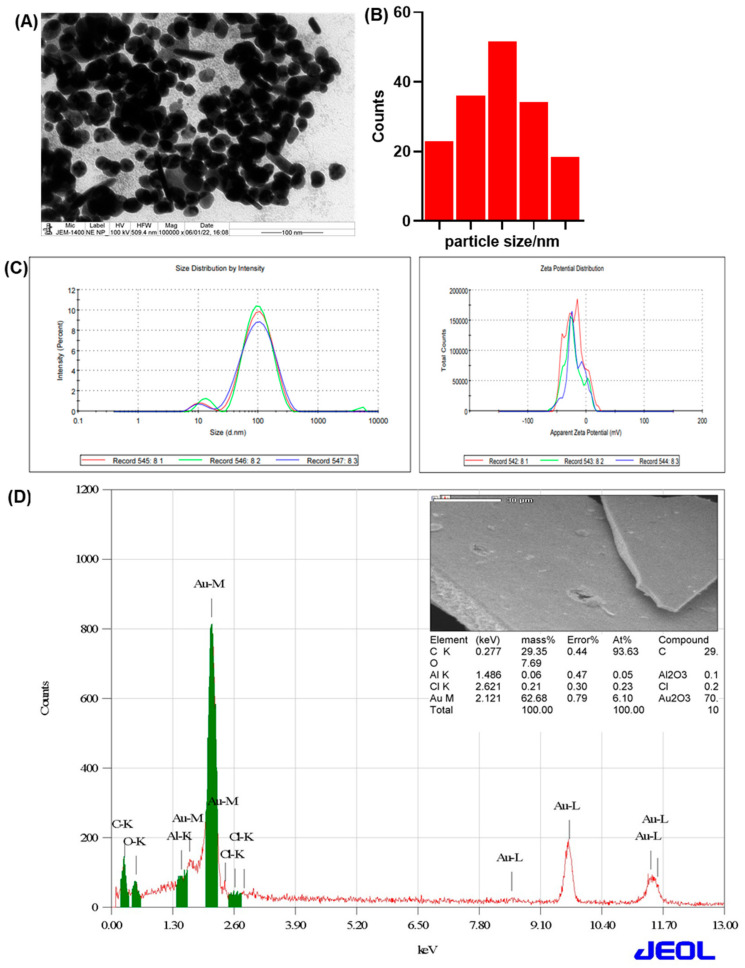
Structural characterization of NC–gold nanoparticles (NC–GNPs): (**A**) transmission electron microscopy (TEM) micrograph of NC–GNPs particle; (**B**) graph showing the size distribution of NC–GNPs; (**C**) graph showing the hydrodynamic size and stability of NC–GNPs analyzed by DLS/zeta potential; and (**D**) scanning electron microscopical (SEM) analysis of NC–GNPs and the EDX profile showed the NC–GNPs capped with Au atom.

**Figure 5 ijms-25-02115-f005:**
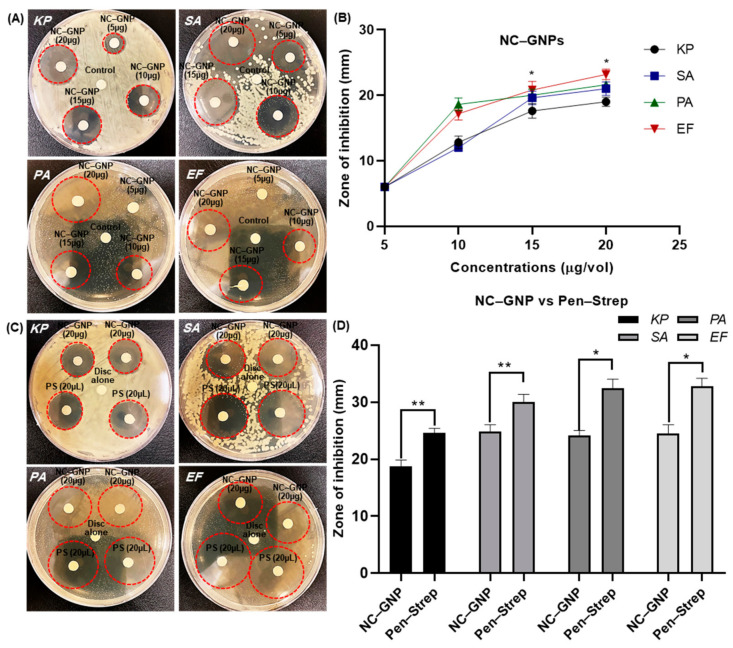
Antimicrobial activity of NC–GNPs optimized with cotton tissue discs: (**A**,**B**) disc diffusion assay at different concentrations of NC–GNPs against *K. pneumoniae*, *S. aureus*, *P. aeruginosa*, and *E. faecalis* bacterial strains; and (**C**,**D**) NC–GNPs (Figure 3(A2)) concentrations optimized versus Pen–Strep solutions in cotton swipe tissue discs. Red dotted circles indicate that the inhibition zone of NC–GNPs and Pen–Strep. The values are represented in millimeters (mm) of bacterial growth inhibition zones. For plating, Luria Bertani (LB), Nutrient, and Brain Heart Infusion (BHI) agar mediums were used. Also, 10 µL of distilled water was added to the control sample disc. * and ** represent significant values compared to control, *p* ≥ 0.05 and 0.01.

**Figure 6 ijms-25-02115-f006:**
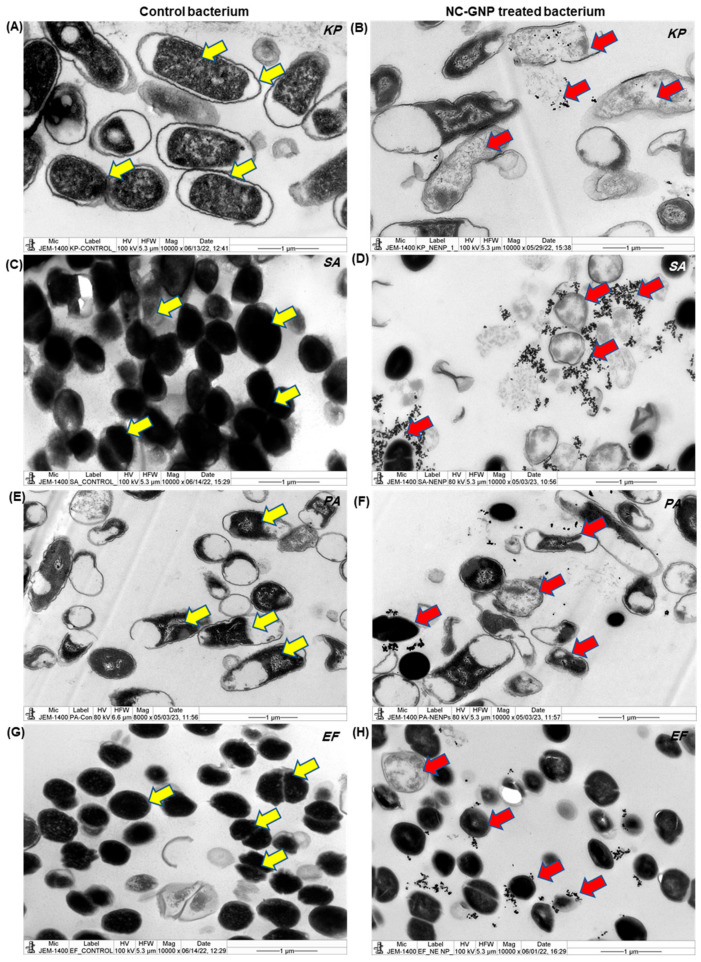
Transmission electron micrograph of control bacteria and bacteria treated with NC–GNPs: (**A**) ultrastructure of control *K. pneumoniae* with a round nucleus and a membrane; (**B**) ultrastructure of *K. pneumoniae* treated with NC–GNP showing degenerative nucleus and plasma membrane; (**C**) ultrastructure of control *S. aureus* with round nucleus and a membrane; (**D**) ultrastructure of *S. aureus* treated with NC–GNP showing vacuolated structures; (**E**,**F**) ultrastructure of *P. aeruginosa* control and NC–GNP treated bacteria; and (**G**,**H**) ultrastructure of *E. faecalis* control and NC–GNP treated bacteria selectively agregated around the cell membranes and degenerating. The images were captured from various locations at a magnification of 10,000× and high magnification (Appendix A). Control bacterial cell walls with nuclei are indicated by yellow arrows, and they migrate routinely towards cell replications. The red arrows represent NC-GNP-treated bacteria that displayed nuclear shrinkage and nuclear disintegration, as well as cellular wall degeneration.

**Table 1 ijms-25-02115-t001:** Major constituents (phytochemicals) present in Nilavembu Choornam polyherbal plants previously reported by Mekala and Murthy, 2020 [31].

S. No.	Botanical Name	Active Phytochemicals
**1.**	*Andrographis paniculata*	Andrographolide, neoandrographolide, 14-deoxyandrographolide, 14-deoxy-12-hydroxy andrographolide, isoandrographolide, andrograpanin β-sitosterol and stigmasterol
**2.**	*Chrysopogon zizanioides* L.	Khusimol, α-vetivone, β-vetivone, zizanal, epizizanal, and nootkatone
**3.**	*Santalum album* L.	α-Santalol, sandalore, β-santalol, (+)-α-santalene, and isobornyl cyclohexanol
**4.**	*Zingiber officinale* R.	Gingerols, shogaols, paradols, quercetin, zingerone, gingerenone-A, 6-dehydrogingerdione β-bisabolene, α-curcumene, zingiberene, α-farnesene, and β-sesquiphellandrene
**5.**	*Piper nigrum* L.	Piperine, sarmentosine, piperamide, piperamine, trichosta, sarmentine, and chavicine
**6.**	*Cyperus rotundus* L.	Patchoulenone, isopatchoulenone, sugenoyl acetate, sugetriol triacetate, sugeibol, kaempferol, luteolin, and quercetin
**7.**	*Mollugo cerviana* L.	C-glycolsyl flavonoids-Orientin and Vitexin
**8.**	*Plectranthus vettiveroides*	Androstan-17-one-3-ethyl-3-hydroxy-(5α)-spathulenol, α-bisabolol, Z-valerenyl acetate, megastigma-4,6(*E*), and 8(Z)-triene
**9.**	*Trichosanthes cucumerina* L.	Cucurbitacin B, E, isocucurbitacin, 23,24-dihydroisocucurbitacin B, E, lycopene, and ascorbic acid

**Table 2 ijms-25-02115-t002:** Composition of Nilavembu Choornam polyherbal decoction [31].

S. No.	Botanical Name	Local Name (English)	Plant Parts
**1.**	*Andrographis paniculata* (Burm. f.) Nees	King of bitters	Whole plant
**2.**	*Chrysopogon zizanioides* (L.) Roberty	Perennial bunchgrass	Root
**3.**	*Santalum album* L.	Sandalwood	Bark
**4.**	*Zingiber officinale* Roscoe	Dry Ginger	Rhizome
**5.**	*Piper nigrum* L.	Black pepper	Fruit
**6.**	*Cyperus rotundus* L.	Purple nutsedge	Root tuber
**7.**	*Mollugo cerviana* (L.) Ser.	Threadstem carpetweed	Whole plant
**8.**	*Plectranthus vettiveroides* (Jacob)	Coleus vettiveroides root	Root
**9.**	*Trichosanthes cucumerina* L.	Snake gourd	Whole plant

## Data Availability

The original contributions presented in this study are included in the manuscript/Appendix A. Further inquiries can be directed to the corresponding author.

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
