# Peer review of "Synergistic Effect of Nilavembu Choornam–Gold Nanoparticles on Antibiotic-Resistant Bacterial Susceptibility and Contact Lens Contamination-Associated Infectious Pathogenicity"

_ijms, 2024, doi:10.3390/ijms25042115_

Round 1
Reviewer 1 Report
Comments and Suggestions for Authors
The comments and suggestions can be found in the attached document.

Reviewer 2 Report
Comments and Suggestions for Authors
The paper presented for review, entitled "Synergistic effect of Nilavembu Choornam-gold nanoparticles on antibiotic-resistant bacterial susceptibility and contact lens contamination-associated infectious pathogenicity", the authors investigated the possibility of using gold nanoparticles obtained by reduction with an herbal extract on antibiotic-resistant bacteria to ensure contact lens sterility. The gold nanoparticles were characterised by appropriate physico-chemical methods (including UV-Vis, SEM-EDX, XRD, Zeta sizer, FTIR and TEM analysis). The authors have made a sound analysis of the literature and I find the paper interesting and valuable.
Below are my comments:
Not all results are discussed and presented fairly. E.g. there is a lack of information on the exact methodology of FTIR, TEM,...). In many places they give the FTIR band frequencies to the second decimal place (for example Line 180-181 "The sharp peak at 1629.05cm-1 was assigned to the C=O, ......" while in others it is rounded to whole numbers. Which is correct? In my opinion giving to the second decimal place does not make sense, the accuracy of spectrophotometers is most often between 2 and 4 cm-1.
The sentence in line 179 "This measures infrared intensity versus wavelength of light." In my opinion it is redundant and adds nothing to the work. If this were to be a definition of the FTIR method-it is too unsuccessful.
It is also worth describing the methodology in detail. How the FTIR spectra were performed (KBr tablets or ATR method).
The first paragraph in the discussion fits more with the introduction.
I think the paper with minor corrections and taking into account comments is suitable for publication.
Reviewer 3 Report
Comments and Suggestions for Authors
Synergistic effect of nilavembu choornam-gold nanoparticles on antibiotic-resistant bacterial susceptibility and contact lens contamination-associated infectious pathogenicity reveals a promising approach in combating contact lens-related infections and enhancing the treatment of resistant microbial pathogenicity. However, in the present form, the paper cannot be accepted for publication, the experimental design is not sufficiently described and the results are not clearly presented, and a major revision is recommended. Other comments are written below:
The names of all bacteria must be written in italics
Table 1 - In my opinion Table 1. Composition of Nilavembu Choornam polyherbal decoction it should be in the material and methods section, not in the section results - determination of bacterial infection and biofilm formation in contact lenses. what does composition of Nilavembu Choornam polyherbal decoction have to do with the subchapter determination of bacterial infection and biofilm formation in contact lenses.
Line 90-95 - What specific mechanisms or factors do you believe contribute to the significant difference in growth, microcolony formation, and biofilm characteristics between antibiotic-resistant bacteria (e.g., K. pneumoniae and S. aureus) and non-resistant bacteria (e.g., P. aeruginosa and E. faecalis) as observed in your study? The growth mode of different bacterial strains depends on the species it is normal to have growth differences.
Figure 2A - According to the general methods used for antibiogram, in order to take into account, the antimicrobial effect of an antibiotic/extract, the diffusion circle must be round.
Table 2. Major constituents (phytochemicals) present in Nilavembu Choornam polyherbal plants – Majors constituents according to who? This table does not seem to be based either on bibliographic sources or on own results
Line 164 - Considering that the synthesis part of the gold nanoparticles was also carried out, that the way in which the synthesis was done should be described more clearly, perhaps even separately from the sub-chapter Characterization of Synthesized NC-GNPs.
What is the difference between Figure 3 A1 and A2?
The description of figure 4 does not match with figure 4, image D cannot be seen on the figure
According to image 4C, the most abundant elements are C and K, not Au. How do you explain that?
Line 216 How the crystallized of the synthesized NC-GNPs was achieved?
Figure 5A, Considering that the inhibition zones are united, how did the author assess the inhibition diameter? Figure 5 The distribution of bacteria on the plates is not uniform
Are there any potential risks or limitations of using NC-GNPs in such a manner, especially considering long-term exposure and environmental impact? the long-term biocompatibility and potential toxicity of GNPs in an ocular environment seem to be outside the scope of the study. Could you discuss any considerations or preliminary findings regarding the safety and biocompatibility of these nanoparticles, especially when they are intended for use in contact lens applications or other direct ocular treatments?
The conclusions do not seem to support the research results, they should be rewritten with a focus on the results obtained, not only on future perspectives
Comments on the Quality of English LanguageI recommend the correction of the text by a native English speaker, in some places the expression is not clear or is incorrect, which makes it difficult to read the text.
Round 2
Reviewer 3 Report
Comments and Suggestions for Authors
I think the authors have made significant improvements to the work, and the work can be published